# Factors associated with neonatal near miss among neonates admitted to public hospitals in dire Dawa administration, Eastern Ethiopia: A case-control study

**Yitagesu Sintayehu**[1]*, **Legesse Abera**[1], **Alekaw Sema**[1], **Yalelet Belay**[1], **Alemu Guta**[1], **Bezabih Amsalu**[1], **Tafese Dejene**[2], **Nigus Kassie**[1], **Teshale Mulatu**[3], **Getahun Tiruye**[3]

1 Department of Midwifery, College of Medicine and Health Sciences, Dire Dawa University, Dire Dawa, Ethiopia, 2 School of Medicine, College of Medicine and Health Sciences, Dire Dawa University, Dire Dawa, Ethiopia, 3 Department of Midwifery, College of Health and Medical Sciences, Haramaya University, Harar, Ethiopia

* yitagesu.sintayehu@gmail.com

**Data Availability Statement:** All relevant data are within the manuscript and its Supporting information files.

## Abstract

### Introduction

The neonatal near-miss cases are subject to factors that are major causes of early neonatal deaths. For every death, more newborns suffer a life-threatening complication. Nearly 98% of neonatal death unduly existed in developing countries. Though there were few prior studies in other regions, they failed in identifying the factors of NNM. Besides, there has been no prior study in the study area. Therefore, this study aimed to assess factors associated with neonatal near-miss.

### Methods

A case-control study was employed on a total of 252 cases and 756 controls using a systematic random sampling technique. Data were collected using pre-tested and interview administered questionnaires adapted from similar studies and medical records from December 2020 –March 2021. Pragmatic and management criteria definition of neonatal near miss were utilized. Epi-Data version 3.1 and SPSS version 23 were used for data entry and analysis respectively. Bivariable and multivariable analyses were done to identify factors associated with a neonatal near-miss by using COR and AOR with a 95% confidence interval. Finally, the statistical significance was declared at a p-value < 0.05.

### Results

There were a response rate of 100% for both cases, and controls. Factors that affects neonatal near miss were non-governmental/private employee (AOR, 1.72[95%CI: 1.037, 2.859]), referral in (AOR, 1.51[95%CI: 1.079, 2.108]), multiple birth (AOR, 2.50[95%CI: 1.387, 4.501]), instrumental assisted delivery (AOR, 4.11[95%CI: 1.681, 10.034]), hypertensive during pregnancy (AOR, 3.32[95%CI: 1.987, 5.530]), and male neonates (AOR, 1.71

**Funding:** Dire Dawa University provided support in the form of per diem for data collectors, supervisors and provided training for the team. No additional external funding was received for this study. The funders had no role in study design, data collection and analysis, decision to publish, or preparation of the manuscript.

**Competing interests:** The authors have declared that no competing interests exist.

**Abbreviations:** ANC, Antenatal Care; APGAR, Appearance Pulse Grimace Activity Respiration; NGO, Non-Governmental Organization; NICU, Neonatal Intensive Care Unit; NNM, Neonatal Near Miss.

[95%CI: 1.230, 2.373]), paternal education of secondary school (AOR, 0.43[95%CI: 0.210, 0.868]) and college/above (AOR, 0.25[95%CI: 0.109, 0.578]), monthly income (1500–3500 birr) (AOR, 0.29[95%CI: 0.105, 0.809]) and >3500 birr (AOR, 0.34[95%CI: 0.124, 0.906]).

## Conclusion

Maternal occupation, paternal education, income, referral, multiple births, mode of delivery, hypertension during pregnancy, and sex of the neonate have identified factors with neonatal near-miss. Better to create job opportunities, improving education, and income generation. Counseling on multiple birth and hypertension, and minimizing instrumental delivery should be done at the health facility level.

## Introduction

A Neonatal near miss (NNM) case refers to a neonate that presents with a severe life-threatening complication during the neonatal period but survives by chance or treatment [1,2]. The term near-miss in pediatrics and neonatology is mostly used in the context of almost adverse events and potential adverse events in the intensive care unit [3].

Worldwide, about 3.6 million neonates are estimated to die in the fififirst 4 weeks of life every year and the majority continue to die at home, uncounted. Out of an estimated, 7.6 million death in under-five children about two-thirds of mortality occurs in the neonatal period and the highest percentage of death happened on the first day and first month of life [4–7]. Likewise, the low- and middle-income countries carry an excessive risk of death and account for three-quarters of neonatal deaths [7,8]. Nearly 98% of neonatal death unduly existed in developing countries and sub-Saharan African countries hold the highest burden of neonatal near misses and deaths [6].

Three major causes of neonatal deaths (infections, complications of preterm birth, and intrapartum-related neonatal deaths or "birth asphyxia") account for more than 80% of all neonatal deaths globally [7,9]. However, the most common causes of neonatal deaths are often preventable [9]. While neonatal mortality is a significant problem in developing countries, for every death there are many more newborns who narrowly miss dying and may suffer long-term consequences as a result [2]. Neonatal near misses, known as miracle babies are newborns who survive from a life-threatening condition [2,10,11].

Neonatal mortality declined in all regions but slower than child mortality. The global neonatal mortality rate dropped from 37 [12,13], deaths per 1,000 live births in 1990 to 19 in 2016 [14,15]. However, there exists a marked variation in the reduction of neonatal mortality rate across regions and countries and the highest rate of mortality rate reaches 28 deaths per 1,000 live births [6].

Neonatal near-miss cases occur more often than neonatal deaths and could enable a more comprehensive analysis of risk factors, short-term outcomes, and prognostic factors in neonates born to mothers with severe obstetric complications [1]. Substantial variations in the mortality among neonates with life-threatening conditions at birth were observed suggesting the intra-hospital quality of care issues. The near-miss concept and indicators provided information that could be useful to evaluate the quality of care and set priorities for further assessments and health care improvement for newborn infants [16].

Furthermore, studying factors associated with near misses has paramount importance to measure the quality of health services, as it is associated with adverse birth outcomes. Even

though certain studies have been conducted in the country and assessed near-miss cases, as far as the investigators' knowledge was concerned there were no studies conducted on factors associated with neonatal near-miss among neonates in the Dire Dawa Administration. Since the study will explain factors associated with neonatal near misses, study results serve as an input for the health bureaus, health offices/departments, local NGOs, and other stakeholders working in Dire Dawa administration in planning and implementation of preventive and intervention strategies to improve maternal and neonatal health. Moreover, the study can also be used as a baseline framework for further studies that will be conducted in similar setups.

## Methods and materials

### Study area and period

The study was conducted in the public hospitals of Dire Dawa administration, Easter Ethiopia, from December 2020 to March 2021. Dire Dawa is found at 515 km to the east of Addis Ababa, the capital city of Ethiopia. It has a projected total population of 506,640; there are 2 public hospitals with 510 expected delivery per month and 15 health centers. These 2 hospitals have both pediatric admission ward and neonatal ICU that are serving for the study area and neighboring areas [17].

### Study design and population

A hospital-based case-control study was employed among neonates who were admitted to public hospitals of the Dire Dawa Administration. Selected neonates who were admitted to the public hospital of Dire Dawa administration and available during the data collection period were included in the study. Mothers or caregivers incapable to give information about the newborn and neonates who were not been found with their mothers (caregivers) were excluded from the study.

### Selection of cases and controls

**Cases (NNM).** Neonates who were admitted to the hospital according to NNM definition, neonates with at least one of the neonatal near miss criteria (with the presence of at least one pragmatic marker and/or management severity criteria) but survived from this condition within the first 28 days were cases [18]. The pragmatic criteria include birth weight <1750gm, an APGAR score <7 at 5th minutes of life and GA < 33 weeks and management severity criteria include: parenteral antibiotic therapy, nasal continuous positive airway pressure (CPAP), any intubation, and phototherapy within 24 hrs. of life, cardiopulmonary resuscitation (CPR), use of vasoactive drugs, anticonvulsants, surfactant, blood products, steroids for the treatment of refractory hypoglycemia, surgery, use of antenatal steroid, use of parenteral nutrition, identification of congenital malformation according to the ICD-10 if considered a near-miss case by another criterion and admission to the NICU [19]. Additionally, data from the record was retrieved to identify the cases and different exposures.

**Controls.** Neonates who were admitted to the post-natal or neonatal ward and identified by a pediatrician or neonatologist or gynecologist or resident as a healthy babies (have no complication indicated for selection of case) were enrolled as a control. For each near-miss case, three controls within the same day of the near-miss event were selected.

### Sample size determination and sampling procedures

The sample size was computed by using sample size determination in an unmatched case-control study in the Epi info7 software Stat Calc. with the assumptions of a 95% level of

confidence, power of 80%, the ratio of cases to control 1:3, and percent of controls exposed 6 and percent of cases with exposure 12.7 taken from others similar study (18). Based on the above assumptions the estimated sample size was 229 cases and 687 controls. After considering the non-response of 10%, the final sample size used for this study was 252 cases and 756 controls. Therefore, the sample size for this study was 1008. A Consecutive sample method was employed to take mothers with neonates.

## Data collection tools, procedures, and quality control

Data were collected by a pre-tested, structured interviewer-administered questionnaire and standard abstraction checklist, which were developed, from different works of literature [18,20]. The questionnaire was prepared in English and then translated to the local languages (Afan Oromo, Amharic, Af Somali) and back-translated to English to ensure the consistency of the thought of the questions. The questionnaire contains information related to socio-demographic characteristics of parents, health care services, obstetrics-related characteristics of the mothers, and a medical card review of the maternal and newborn condition. Data were collected by 6 midwives who have experience in maternal and neonatal care and non-staffs in the study setting.

To ensure the data quality, the tool was translated into the local languages. Additionally, pre-testing was done on 13 cases and 38 controls during the data collector training and modified based on the findings accordingly. The questionnaires were checked for completeness and consistency. Data collection was supervised by the principal and co-investigators regularly throughout the data collection period. Finally, double data entry was done to minimize errors during data entry.

## Operational definition of variables

**Neonate**: a newborn age period that is birth to 28 days.

**Neonatal near miss**: defined as the presence of at least one pragmatic marker or management severity criteria [19].

**Pragmatic markers criteria**: It is the severity of a criterion that is used to classify a neonate as a neonatal near miss. It includes birth weight < 1750 g, an APGAR score < 7 at 5 min, and Gestational Age < 33 weeks [19].

**Management severity criteria**: It is a criterion based on the management base. It includes parenteral antibiotic therapy, nasal continuous positive airway pressure (CPAP), any intubation, and phototherapy within 24 hrs. of life, cardiopulmonary resuscitation (CPR), use of vasoactive drugs, anticonvulsants, surfactants, blood products, steroids for the treatment of refractory hypoglycemia, surgery, use of antenatal steroid, use of parenteral nutrition, identification of congenital malformation according to the ICD-10 if considered a near-miss case by another criterion and admission to the NICU [19].

**Medical complications during pregnancy**: include diabetes mellitus, hypertension, tuberculosis, cardiac disease, malaria, anemia, and some other related medical conditions observed during this pregnancy [19].

## Data analysis

The Collected data were checked for completeness, coded, entered, and cleaned using Epi-Data version 3.1, and exported to SPSS version 23 for analysis. Descriptive statistics were used to describe the frequency distribution of each of the variables mentioned earlier. The association between the outcome variables (i.e. neonatal near-miss) and independent variables was analyzed using a binary logistic regression model. Covariates having a p-value of 0.25 or less

will be retained and entered into the multivariable logistic regression analysis using forward stepwise approach methods. The Hosmer and Lemeshow goodness-of-fit test were used to assess whether the necessary assumptions for the application of multiple logistic regression were fulfilled and a p-value > 0.05 was considered a good fit. The result was presented as adjusted odds ratios with 95% confidence intervals. A p-value<0.05 was considered for significantly associated factors with the outcome variable.

## Ethical considerations

Ethical clearance was first sought from the Research and Ethical Review Committee of the college of medicine and health sciences, Dire Dawa University with ethics reference number, ም/ማ/ኣ/ም/ፐ300/879/2013. Next, a permission letter was collected from the respective Health Bureaus and communicated with hospitals. After discussing the issue of confidentiality data collectors were taken informed, voluntary, written, and signed consent from the hospital head and participants or guardians of the minors after taking assent from the minors before the start of data collection. Participants were informed that they have the full right to refuse or discontinue participating in the study and as there would not be any marker to identify any participants' privacy would be kept during interviews and responses would be kept confidential. All personal information was de-identified and kept separately, so every effort was made to maintain confidentiality throughout the study period and afterward. Furthermore, this study was conducted in accordance with the Declaration of Helsinki.

## Result

### Sociodemographic characteristics of neonates mothers

A total of, 252 cases and 756 controls were involved in the study and provide a response rate of 100% for both cases, and controls. The mean age and a standard deviation of neonate's mother were 25.90 ± 5.19 for the whole study participants and 25.97 ± 5.12 for cases and 25.87 ± 5.21 for controls.

One hundred ninety-five (77.4%) of the neonate's mothers for cases and 664 (87.6%) for controls were urban residents. The majority of participants, 62.7% in cases and 55% in controls are housewives. In the case of marital status, only 1.2% in cases and 1.1% in controls are currently single. Regarding maternal educational status, 9.9% of cases and 15.3% of controls attended college and above (Table 1).

### Maternal and child health, and obstetric factors

Among participants, in 85.3% of cases and 89% of controls pregnancies are planned and wanted. Of multi-para mothers', 38.9% of cases and 40.9% of controls have birth intervals of less than 24 months. ANC follow-up was attended in 94% of cases and 97.1% of controls at least one time. Of the total neonates, 64.7% are males from cases and 51.7% from controls. Regarding mode of delivery, 63.5% of cases and 64% of controls were delivered by spontaneous vaginal delivery (Table 2).

### Presences of current or history of maternal obstetric complication

In the current study, no one has a stillbirth history in cases, while 1.6% of controls have it. In the case of abortion, among participants, 12.3% of cases and 11.6% of controls have at least one abortion history. About 7.1% of cases and 4.1% of controls were multiple births. Anemia was found in 15.5% of cases and 20.5% of controls (Fig 1).

## Neonatal near miss diagnostic criteria distribution

According to our findings, out of neonatal near-miss cases, 93.7% of them were admitted to NICU followed by an APGAR score of less than 7 (64.3%). Additionally, 22.8% of neonatal near-miss cases were less than 1750gm in weight and 4.4% of cases were given anticonvulsant drugs (Fig 2).

## Factors associated with the neonatal near miss

In bivariable logistic regression, age of the mother, residence, maternal educational status, maternal occupation, paternal educational status, family monthly income, number of

**Table 1. Sociodemographic characteristics of mothers whose neonates were admitted to public hospitals of Dire Dawa Administrative, Eastern Ethiopia, 2021.**

| Variables | Total N (%) | Cases (n = 252) | Controls (n = 756) |
|---|---|---|---|
| **Age** | | | |
| 15–24 | 412(40.9) | 110(43.7) | 302(40.0) |
| 25–34 | 513(50.9) | 118(46.8) | 395(52.2) |
| ≥35 | 83(8.2) | 24(9.5) | 59(7.8) |
| **Resident** | | | |
| Urban | 859(85.2) | 195(77.4) | 664(87.8) |
| Rural | 149(14.8) | 57(22.6) | 92(12.2) |
| **Maternal educational status** | | | |
| No formal education | 177(17.6) | 76(30.2) | 101(13.4) |
| Primary (1–8) | 360(35.7) | 92(36.5) | 268(35.4) |
| Secondary (9–12) | 330(32.7) | 59(23.4) | 271(35.8) |
| College and above | 141(14.0) | 25(9.9) | 116(15.3) |
| **Maternal occupation** | | | |
| Housewife | 574(56.9) | 158(62.7) | 416(55.0) |
| Merchant | 95(9.4) | 18(7.1) | 77(10.2) |
| Government employer | 157(15.6) | 31(12.3) | 126(16.7) |
| Non-governmental/private | 157(15.6) | 39(15.5) | 118(15.6) |
| Daily laborer | 25(2.5) | 6(2.4) | 19(2.5) |
| **Marital status** | | | |
| Currently Single | 11(1.1) | 3(1.2) | 8(1.1) |
| Currently Married | 997(98.9) | 249(98.8) | 748(98.9) |
| Paternal education status (n = 997) | | | |
| No formal education | 118(11.8) | 52(20.9) | 66(8.8) |
| Primary (1–8) | 197(19.8) | 74(29.7) | 123(16.4) |
| Secondary (9–12) | 389(39.0) | 83(33.3) | 306(40.9) |
| College and above | 293(29.4) | 40(16.1) | 253(33.8) |
| **Religion** | | | |
| Orthodox | 285(28.3) | 67(26.6) | 218(28.8) |
| Muslim | 658(65.3) | 169(67.1) | 489(64.7) |
| Protestant | 44(4.4) | 13(5.2) | 31(4.1) |
| Catholic | 21(2.1) | 3(1.2) | 18(2.4) |
| **Family monthly income** | | | |
| <1500 ETB | 21(2.1) | 10(4.0) | 11(1.5) |
| 1500–3500 ETB | 236(23.4) | 83(32.9) | 153(20.2) |
| >3500 ETB | 751(74.5) | 159(63.1) | 592(78.3) |

**Key: ETB**, Ethiopian Birr.

**Table 2. Maternal and child health, and health care service characteristics of participants in public hospitals of Dire Dawa Administrative, Eastern Ethiopia, 2021.**

| Variables | Total N (%) | Cases (n = 252) | Controls (n = 756) |
|---|---|---|---|
| **Pregnancy status** | | | |
| Planned and wanted | 888(88.1) | 215(85.3) | 673(89.0) |
| Unplanned but wanted | 120(11.9) | 37(14.7) | 83(11.0) |
| **Gravidity** | | | |
| Primi-gravida | 365(36.2) | 103(40.9) | 262(34.7) |
| Multi-gravida | 532(52.8) | 112(44.4) | 420(55.6) |
| Grand multi-gravida | 111(11.0) | 37(14.7) | 74(9.8) |
| **Birth interval (in months) (643)** | | | |
| <24 | 260(40.4) | 58(38.9) | 202(40.9) |
| 24–60 | 361(56.1) | 87(58.4) | 274(55.5) |
| >60 | 22(3.4) | 4(2.7) | 18(3.6) |
| **Antenatal care visit** | | | |
| Yes | 971(96.3) | 237(94.0) | 734(97.1) |
| No | 37(3.7) | 15(6.0) | 22(2.9) |
| **Number of antenatal care visits** | | | |
| No visit | 37(3.7) | 15(6.0) | 22(2.9) |
| 1–3 visit | 475(47.1) | 127(50.4) | 348(46.0) |
| ≥4 visit | 496(49.2) | 110(43.7) | 386(51.1) |
| **Referred in** | | | |
| Yes | 578(57.3) | 173(68.7) | 405(53.6) |
| No | 430(42.7) | 79(31.3) | 351(46.4) |
| **Labor status** | | | |
| Spontaneous | 839(83.2) | 206(81.7) | 633(83.7) |
| Induced | 169(16.8) | 46(18.3) | 123(16.3) |
| **Duration of labor** | | | |
| Elective cesarean section | 87(8.6) | 28(11.1) | 59(7.8) |
| Precipitative | 55(5.5) | 14(5.6) | 41(5.4) |
| Normal duration | 824(81.7) | 195(77.4) | 629(83.2) |
| Prolonged | 42(4.2) | 15(6.0) | 27(3.6) |
| **Mode of delivery** | | | |
| Spontaneous vaginal delivery | 644(63.9) | 160(63.5) | 484(64.0) |
| Instrumental delivery | 24(2.4) | 11(4.4) | 13(1.7) |
| Cesarean section delivery | 340(33.7) | 81(32.1) | 259(34.3) |
| **Presentation** | | | |
| Cephalic | 954(94.6) | 236(93.7) | 718(95.0) |
| Non-cephalic | 54(5.4) | 16(6.3) | 38(5.0) |
| **Sex of neonate** | | | |
| Male | 554(55.0) | 163(64.7) | 391(51.7) |
| Female | 454(45.0) | 89(35.3) | 365(48.3) |
| **Neonatal birth trauma** | | | |
| Yes | 9(0.9) | 8(3.2) | 1(0.1) |
| No | 999(99.1) | 244(96.8) | 755(99.9) |
| **NRFHB** | | | |
| Yes | 184(18.3) | 182(72.2) | 2(0.3) |
| No | 824(81.7) | 70(27.8) | 754(99.7) |

**Keys: NRFHB**, None Reassuring Fetal Heart Beat.

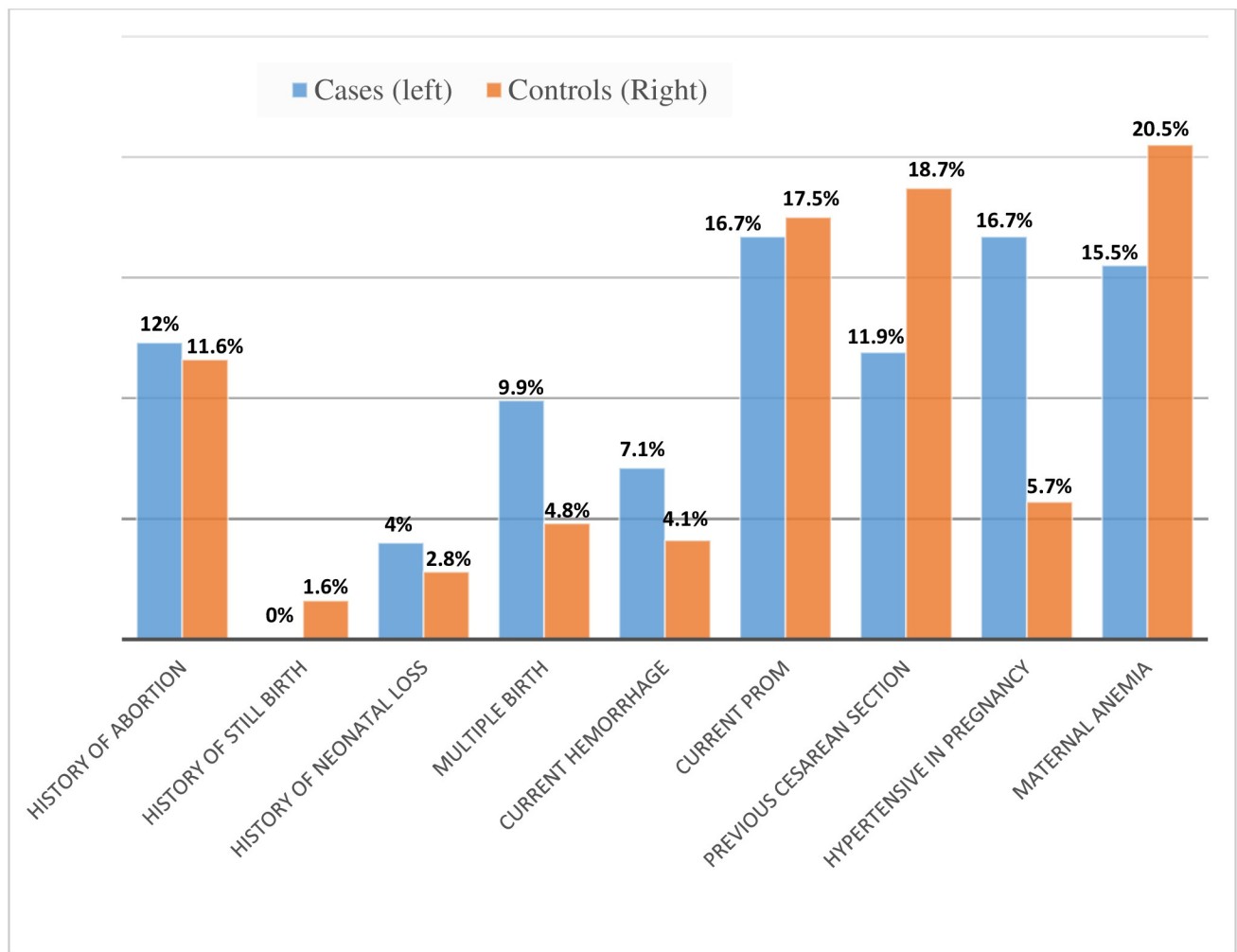

**Fig 1. Presences of current or history of maternal obstetric complication of participants in public hospitals of Dire Dawa Administrative, Eastern Ethiopia, 2021.**

pregnancies, having ANC follow-up, referral in, multiple births, mode of delivery, having a previous cesarean delivery, presence of hypertension in pregnancy, and sex of the neonate was eligible for multivariable analysis.

In the multivariable logistic analysis, maternal occupation, paternal educational status, family monthly income, referral in, multiple births, mode of delivery, hypertension during pregnancy, and sex of the neonate were significantly associated with NNM.

Mothers who were non-governmental/private employees were 2 times more likely to have NNM cases compared to being housewives (AOR, 1.72[95%CI: 1.037, 2.859]). Having paternal education of secondary school (9–12) and college/above were less likely to develop NNM cases compared to those with no formal education (AOR, 0.43[95%CI: 0.210, 0.868]) and (AOR, 0.25[95%CI: 0.109, 0.578]) respectively. Having monthly family income 1500–3500 Ethiopian birr and >3500 Ethiopian birrs were less likely to have NNM cases compared to those have less than 1500 Ethiopian birr monthly income (AOR, 0.29[95%CI: 0.105, 0.809]) and (AOR, 0.34[95%CI: 0.124, 0.906]) respectively.

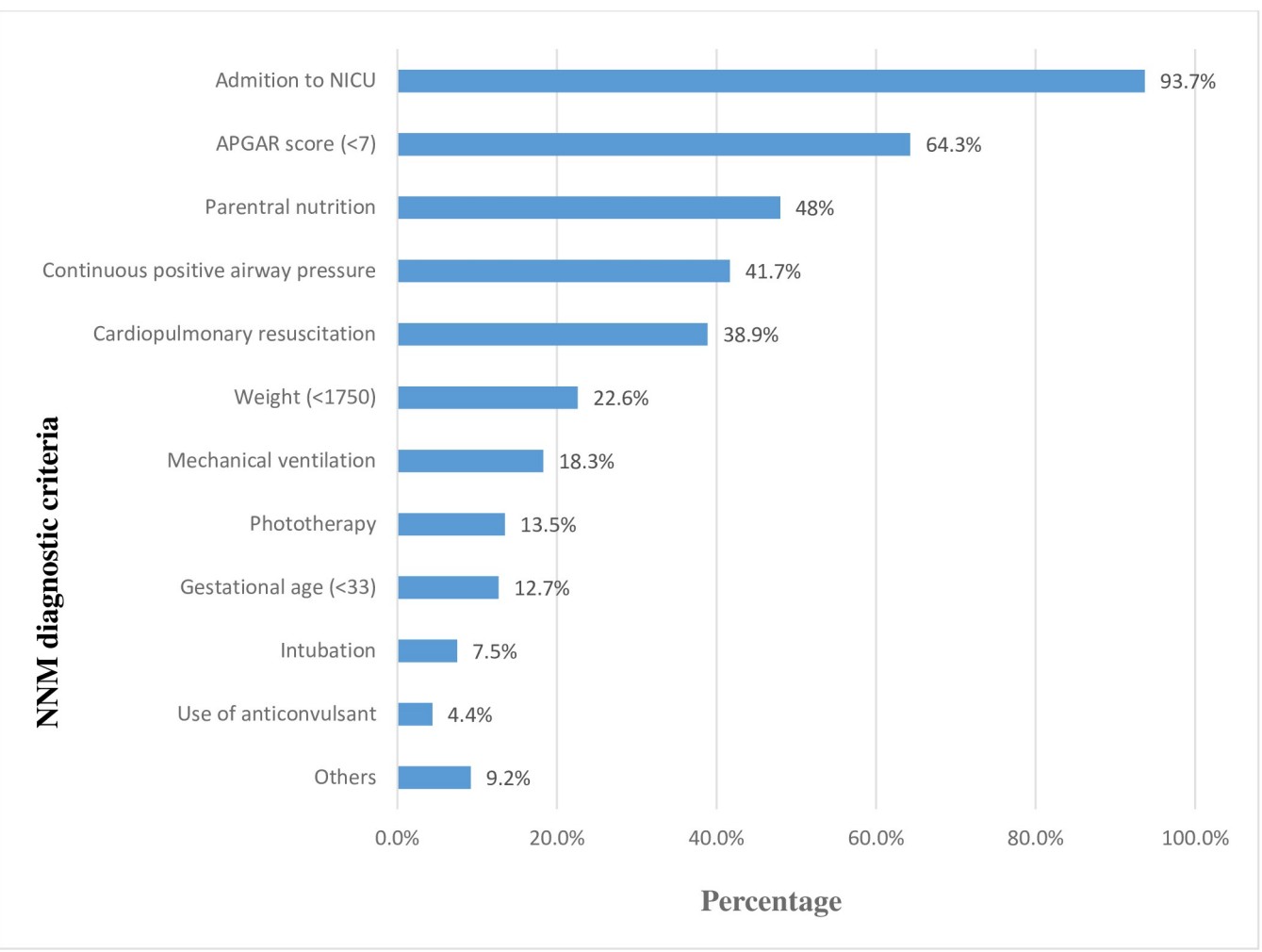

**Fig 2. Neonatal near miss diagnostic Criteria distribution among the case of the participants in public hospitals of Dire Dawa Administrative, Eastern Ethiopia, 2021.**

Mothers of the neonates referred from other facilities were 1.5 times more likely to have NNM cases than those not referred (AOR, 1.51[95%CI: 1.079, 2.108]). Having multiple births was 2.5 times more likely to have NNM cases than having a singleton birth (AOR, 2.50[95%CI: 1.387, 4.501]). Giving birth through instrumental assisted delivery was 4 times more likely to have NNM cases than giving birth through spontaneous vaginal delivery (AOR, 4.11[95%CI: 1.681, 10.034]). The odds of NNM were 3.32 among mothers who have hypertensive during pregnancy as compared to those with no hypertensive during pregnancy (AOR, 3.32[95%CI: 1.987, 5.530]). Furthermore, being male neonates is about 2 times more likely to be NNM cases compared to female neonates (AOR, 1.71[95%CI: 1.230, 2.373]) (Table 3).

## Discussion

Studying factors affecting neonatal near-miss cases is crucial to tackling the cause of the neonatal problems. Therefore, this study showed the most important factors affecting near-miss cases in Dire Dawa Administrative public hospitals. The finding of the present study in the multivariable logistic analysis showed that maternal occupation, paternal educational status,

**Table 3. Multivariable analysis of the factors associated with neonatal near-miss among neonates admitted in public hospitals of Dire Dawa Administrative, Eastern Ethiopia, 2021.**

| Variables | Cases (n = 252) | Controls (n = 756) | COR (95%) | AOR (95%) | p-value |
|---|---|---|---|---|---|
| **Age** | | | | | |
| 15–24 | 110 | 302 | 1 | | |
| 25–34 | 118 | 395 | **0.82(0.608, 1.107)** | 0.94(0.649, 1.353) | 0.729 |
| ≥35 | 24 | 59 | 1.12(0.662, 1.883) | 1.02(0.521, 1.992) | 0.957 |
| **Resident** | | | | | |
| Urban | 195 | 664 | 1 | 1 | |
| Rural | 57 | 92 | **2.11(1.462, 3.045)** | 1.12(0.714, 1.748) | 0.627 |
| **Maternal educational status** | | | | | |
| No formal education | 76 | 101 | 1 | 1 | |
| Primary (1–8) | 92 | 268 | **0.46(0.312, 0.667)** | 0.73(0.415, 1.273) | 0.264 |
| Secondary (9–12) | 59 | 271 | **0.29(0.192, 0.436)** | 0.57(0.282, 1.151) | 0.116 |
| College and above | 25 | 116 | **0.29(0.169, 0.484)** | 0.69(0.282, 1.690) | 0.417 |
| **Maternal occupation** | | | | | |
| Housewife | 158 | 416 | 1 | 1 | |
| Merchant | 18 | 77 | **0.62(0.357, 1.061)** | 1.16(0.624, 2.145) | 0.644 |
| Government employer | 31 | 126 | **0.65(0.420, 0.999)** | 1.70(0.891, 3.226) | 0.108 |
| Non-governmental/private | 39 | 118 | 0.87(0.580, 1.306) | **1.72(1.037, 2.859)** | **0.036***  |
| Daily laborer | 6 | 19 | 0.83(0.326, 2.120) | 1.03(0.362, 2.931) | 0.956 |
| **Paternal education status (n = 997)** | | | | | |
| No formal education | 52 | 66 | 1 | 1 | |
| Primary (1–8) | 74 | 123 | 0.76(0.480, 1.214) | 0.94(0.507, 1.738) | 0.841 |
| Secondary (9–12) | 83 | 306 | **0.34(0.222, 0.533)** | **0.43(0.210, 0.868)** | **0.019***  |
| College and above | 40 | 253 | **0.20(0.123, 0.329)** | **0.25(0.109, 0.578)** | **0.001***  |
| **Family monthly income** | | | | | |
| <1500 Ethiopian Birr | 10 | 11 | 1 | 1 | |
| 1500–3500 Ethiopian Birr | 83 | 153 | 0.60(0.243, 1.463) | **0.29(0.105, 0.809)** | **0.018***  |
| >3500 Ethiopian Birr | 159 | 592 | **0.30(0.123, 0.708)** | **0.34(0.124, 0.906)** | **0.031***  |
| **Gravidity** | | | | | |
| Primi-gravida | 103 | 262 | 1 | 1 | |
| Multi-gravida | 112 | 420 | **0.68(0.498, 0.924)** | 0.77(0.516, 1.134) | 0.182 |
| Grand multi-gravida | 37 | 74 | 1.27(0.806, 2.006) | 0.94(0.509, 1.746) | 0.852 |
| **ANC visit** | | | | | |
| Yes | 237 | 734 | 1 | 1 | |
| No | 15 | 22 | **2.11(1.078, 4.137)** | 1.32(0.611, 2.832) | 0.483 |
| **Referred in** | | | | | |
| Yes | 173 | 405 | **1.90(1.403, 2.567)** | **1.51(1.079, 2.108)** | **0.016***  |
| No | 79 | 351 | 1 | 1 | |
| **Multiple births** | | | | | |
| Yes | 25 | 36 | **2.20(1.294, 3.748)** | **2.50(1.387, 4.501)** | **0.002***  |
| No | 227 | 720 | 1 | 1 | |
| **Mode of delivery** | | | | | |
| SVD | 160 | 484 | 1 | 1 | |
| Instrumental | 11 | 13 | **2.56(1.124, 5.827)** | **4.11(1.681, 10.034)** | **0.002***  |
| Cesarean section | 81 | 259 | 0.95(0.696, 1.286) | 1.03(0.703, 1.504) | 0.887 |
| **Previous cesarean section** | | | | | |
| Yes | 30 | 141 | 1 | 1 | |

*(Continued)*

**Table 3.** (Continued)

| Variables | Cases (n = 252) | Controls (n = 756) | COR (95%) | AOR (95%) | p-value |
|---|---|---|---|---|---|
| No | 222 | 615 | **1.70(1.112, 2.590)** | 1.20(0.707, 2.051) | 0.494 |
| **Hypertensive during pregnancy** | | | | | |
| Yes | 42 | 43 | **3.32(2.110, 5.212)** | **3.32(1.987, 5.530)** | **0.001**[*] |
| No | 210 | 713 | 1 | 1 | |
| **Sex of neonate** | | | | | |
| Male | 163 | 391 | **1.71(1.273, 2.297)** | **1.71(1.230, 2.373)** | **0.001**[*] |
| Female | 89 | 365 | 1 | 1 | |

[*]significantly associated at p-value <0.05;

**ANC**, Antenatal Care; **SVD**, Spontaneous Vaginal Delivery.

family monthly income, referral in, multiple births, mode of delivery, hypertension during pregnancy, and sex of the neonate were significantly associated with NNM.

In the current study, maternal occupation, non-governmental and private employment was significantly associated with a neonatal near miss. This is in line with the study conducted in Hawasa city [21]. This might be due to the adverse effects of occupational stress on fetal growth and development. This is supported by those mothers in hard-working conditions who might be a risk factor for adverse birth outcomes when compared to mothers of housewives. Other study evidence for the adverse effects of occupational stress on fetal growth and development also supports this [13].

Neonates delivered from mothers referred from other health facilities had higher odds of a neonatal near miss than those not referred in. This finding was in line with the studies conducted in Uganda and Gurage Zone [22,23]. This might be due to the delayed/absence of referral on time to the next facility and interruption of care given to the mothers until they arrive at the next facility. Having multiple births is a factor significantly associated with neonatal near misses. This is similar to the study finding in South Africa [1]. This might be due to that multiple pregnancies are expected to affect fetal outcomes in terms of underweight, preterm delivery, and asphyxia.

Mode of delivery is significantly associated with neonatal near misses. Mothers delivered through instrumental delivery were more likely to develop neonatal near-misses than spontaneous vaginal delivery. This finding was similar to studies conducted in different settings, in Brazil, Johannesburg, Southeast Brazil, Hawasa, northeast Brazil, Gamo and Gofa zones, Uganda, Ambo, Gurage Zone, Jima, Thailand, Morocco, and Ethiopia [10,12,18,20–22,24–30]. This might be because most of the neonates indicated for instrumental delivery are due to abnormal labor progress, which affects the fetal outcomes, and maybe the professionals are not skilled enough, maybe the mothers are frightened by the indications that happen later in worse conditions for the fetuses. A neonate delivered from mothers having hypertension during pregnancy was more likely to face neonatal near misses. This finding was in line with studies conducted in a different studies, in Uganda, Gondar, Southeast Brazil, and Jima [22–24,27]. This might be because neonates delivered from hypertensive mothers may be affected by the impact of the maternal hypertensive and its management (drugs). Additionally, this might be due to hypertension during pregnancy may cause complications to fetuses during intrauterine life like intrauterine growth restriction and in extrauterine life such as preterm delivery which is more likely to be LBW and also causes birth asphyxia [31]. However, inconsistent with the Hawasa city study. This might be due to the sample size difference, which is our study has almost two times higher than their study.

Another factor significantly associated in this study was paternal educational status, which was having paternal education of secondary school and college/above had a significant association. This variable was tested and failed to associate with other studies (10, 27). This association might be due to that having better education in husbands leads to better awareness of maternal health care access and better income to afford the care to get a better follow-up to identify risk factors.

Family monthly income was a factor significantly associated with NNM, which is having a better family monthly income was less likely to have NNM. This might be having a better income leads to accessing better maternal health care services in a different area or nearby their residency even in private facilities since they can afford it. This finding is supported by the study conducted in south Ethiopia [32].

Male neonates had a significant factor for having neonates with a near miss as indicated in this study. Neonatal hypoglycemia and immediate neurological complications were significantly more frequent in males. For term small for gestational age, low 5-min APGAR scores (<7) at 39–40 weeks were higher for males compared with females, as was hypoglycemia [33].

This study showed the gaps in previous studies to show the factors that could affect neonatal near-miss cases. Therefore, this study identified some new variables that were not identified by other studies, such as paternal educational status and being a male neonate to fill the previously mentioned gaps. However, this study is unable to follow neonates with near misses until the end of 28 days of life to see their outcomes, consider seasonality as a risk factor of NNM, and did not incorporate some of the variables that are needed to be addressed in the community, such as wealth index, nutritional status, and cultural aspects.

## Conclusion

Maternal occupation, paternal education, income, referral in, multiple births, mode of delivery, hypertension during pregnancy, and sex of the neonate have been identified factors with neonatal risk factors. The respective body needs to generate income, by creating job opportunities, improving education through giving educational scholars, early screening and managing multiple pregnancies and pregnancy-induced hypertension, minimizing instrumental delivery through careful (expertise) assessment of the indications, creating a system of referral decrease or early referral and give focus for male newborns.

## Data sharing statement

All related data has been presented within the manuscript. The data set supporting the conclusions of this article is available from the corresponding Author (Yitagesu Sintayehu) upon reasonable request.

## Supporting information

**S1 Dataset. The dataset from which the results of the study were produced (SPSS file).** (SAV)

**S1 Questionnaire. The data collection tool (questionnaire and checklist) in English.** (DOCX)

**S1 Ethics letter. The letter of ethical clearance taken to conduct this study.** (PDF)

## Acknowledgments

We are pleased to thank Dire Dawa University for the ethical review to conduct this study. Our gratitude also extends to the Dire Dawa administration health bureau for providing all the necessary data on the target population, which is important for this research proposal. Secondly, we would like to thank all data collectors and supervisors for their valuable contribution to collecting and facilitating data collection. In addition, we would like to thank study participants to participate in this study.

## Author Contributions

**Conceptualization:** Yitagesu Sintayehu, Legesse Abera, Alekaw Sema, Yalelet Belay, Teshale Mulatu.

**Data curation:** Yitagesu Sintayehu, Legesse Abera, Alekaw Sema, Bezabih Amsalu, Tafese Dejene.

**Formal analysis:** Yitagesu Sintayehu, Legesse Abera, Alemu Guta, Bezabih Amsalu, Teshale Mulatu, Getahun Tiruye.

**Funding acquisition:** Yitagesu Sintayehu.

**Investigation:** Yitagesu Sintayehu, Legesse Abera, Bezabih Amsalu, Nigus Kassie, Teshale Mulatu, Getahun Tiruye.

**Methodology:** Yitagesu Sintayehu, Legesse Abera, Alekaw Sema, Nigus Kassie.

**Project administration:** Yitagesu Sintayehu, Legesse Abera, Tafese Dejene.

**Resources:** Yitagesu Sintayehu.

**Software:** Yitagesu Sintayehu, Bezabih Amsalu, Nigus Kassie.

**Supervision:** Yitagesu Sintayehu, Legesse Abera, Yalelet Belay, Alemu Guta, Getahun Tiruye.

**Validation:** Yitagesu Sintayehu, Legesse Abera, Alekaw Sema, Yalelet Belay, Alemu Guta, Bezabih Amsalu, Tafese Dejene, Nigus Kassie, Teshale Mulatu.

**Visualization:** Yitagesu Sintayehu, Teshale Mulatu.

**Writing – original draft:** Yitagesu Sintayehu, Legesse Abera, Getahun Tiruye.

**Writing – review & editing:** Legesse Abera, Alekaw Sema, Yalelet Belay, Alemu Guta, Bezabih Amsalu, Tafese Dejene, Nigus Kassie, Teshale Mulatu, Getahun Tiruye.

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
