## [Decision Letter · Decision Letter 0]

19 May 2022

PONE-D-21-33319

FACTORS ASSOCIATED WITH NEONATAL NEAR MISS AMONG NEONATES ADMITTED TO PUBLIC HOSPITALS IN DIRE DAWA ADMINISTRATION, EASTERN ETHIOPIA: A CASE-CONTROL STUDY

PLOS ONE

Dear Dr. Sintayehu,

Thank you for submitting your manuscript to PLOS ONE. After careful consideration, we feel that it has merit but does not fully meet PLOS ONE’s publication criteria as it currently stands. Therefore, we invite you to submit a revised version of the manuscript that addresses the points raised during the review process.

The manuscript and the reviewers’ comments were carefully evaluated. The Reviewers appreciated the manuscript; however, they highlighted different points of concern that need major revision before considering the manuscript for publication. Suggested revisions and highlighted study limits are in detail reported in the Reviewers’ comments. Moreover, a language revision is recommended.

We look forward to receiving your revised manuscript.

Kind regards,

Simone Garzon

Academic Editor

PLOS ONE

**Journal requirements:**

Reviewers' comments:

Reviewer's Responses to Questions

**Comments to the Author**

1. Is the manuscript technically sound, and do the data support the conclusions?

Reviewer #1: Yes

Reviewer #2: Yes

Reviewer #3: Yes

Reviewer #4: Yes

2. Has the statistical analysis been performed appropriately and rigorously? 

Reviewer #1: Yes

Reviewer #2: Yes

Reviewer #3: Yes

Reviewer #4: Yes

3. Have the authors made all data underlying the findings in their manuscript fully available?

Reviewer #1: Yes

Reviewer #2: No

Reviewer #3: Yes

Reviewer #4: Yes

4. Is the manuscript presented in an intelligible fashion and written in standard English?

Reviewer #1: Yes

Reviewer #2: No

Reviewer #3: Yes

Reviewer #4: No

5. Review Comments to the Author

Reviewer #1: General comments

Dear authors on your scholarly work; you have brought an important study problem with good findings that have public health importance in the area of practice. Moreover, the manuscript has been written in good English. However, it still needs improvement for mainly grammar usage so that its readership becomes increased if published.

Some of the language use errors which the authors made consistently include.

Eg.

a. Card review……… instead of chart review

b. Bivariate and multivariate ….. instead of bivariable and multivariable

Specific comments

1. Background of the abstract doesn’t clearly show the existing numerical burden of the problem in Dire Dawa town or other regional states of Ethiopia even. Generally, burden of neonatal near miss should be stated numerically followed by the objectives showing the research gap the authors would like to address.

2. The fourth sentence of the background that reads as ‘However, little studies were done in other areas, but they failed in identifying the proximate factors and not done in Dire Dawa Administration’ should be rewritten as: Though there were few prior studies in other regions, they failed in identifying the proximate factors. Besides, there has been no prior study in the study area.

3. Methods of abstract should include sampling technique, measurement of neonatal nearmiss and type of data collection tool (adapted or adopted)

4. Result of abstract: please include response rate in the first sentence of the result section.

Introduction

5. Well written except the need for further synthesis of redundant concepts.

Methods

6. Data collection tools, procedures, and quality control: It would be more self explanatory and easily understandable if the authors showed pictorial presentation (flow chart) of the sampling procedure to reach a response rate of 100%. Please upload your data collection tools as additional file than annexing it in the manuscript. Kindly state the exact number of cases and controls rather than mentioning the routine statistics i.e. 5% of the sample size.

7. Ethical consideration: kindly have a separate upload of the ethical letter as an additional file. Moreover, what beneficent actions did the authors provide the mothers and the facilities in return for the interviews and chart reviews?

8. Results

How were you able to measure monthly income because you haven’t planned principal component analysis?

9. Conclusion

Well supported by the findings

Reviewer #2: Title : Factors associated with neonatal near miss among neonates admitted to public hospitals in Dire Dawa administration, Eastern Ethiopia: a case-control study

Thank you for the opportunity for giving me to review the above title. I found the topic is interesting and has implications for low-income countries. However, the manuscript needs extensive language revisions.

Abstract

- Line 21- The near-miss cases are subject to factors…… can you say it neonatal near miss instead of near-miss since there are other miss e.g maternal near miss.

- I do not think by chance is appropriate word to use. Please get rid of the word ‘chance’

- Line 29 – say medical record instead of ‘card’

- Line 30 – Multivariate is not the correct word , replace it by ‘multivariable analysis’

- Line 31-32 -the authors need to re-write the sentence

- Line 33 – no need of mentioning in the abstract about p-values. Please take it out from the abstract.

- Line 35 -39 – can you make the C.Interval to two digits rather than three.

- I don’t recommend stating adjusted ratios and CI in the abstract. It would be better if you could mention the factors associated with NNM without AOR.

- Line 41 to 43 -seems repetition of results. I would recommend to state in other ways.

- Better to create job opportunities, improving education, income generation. How? In any particular way?

- …… minimizing instrumental delivery. How?

Background

- I would start the background with paragraph 4 ‘Worldwide, about 3.6 million neonates are estimated to die in the first 4 weeks of life every year and the majority continue to die at home, uncounted.’

- The authors need to change the order of the paragraphs in order to make it easy for readers

- It needs language revision

Methods

- Can you add more information about study facility such as availability of ICU units , paediatrics admission and number of deliveries per year ?

- Line 111 -12 – not clear

- Line 116 – 120 – It is long sentence. Can you make it two sentences?

- Line 121 – 127 – Repetition of background information. It would be good if the authors mention the criteria of NNM only in methods part. Please take it out from the background.

- Line 137- please include reference.

- Line 148- ‘Data were collected by 6 midwives who have experience in maternal and neonatal care’. Are the midwives hospital staff?

- Line 150 -153- Please make it two sentence.

- Can you include reference number for ethics approval?

- ‘This might be due to that the adverse effects of occupational stress on fetal

growth and development. Do you have reference for this ? Can we say private /non-governmental employment increase the risk of adverse outcomes?

- Line 302 _ having paternal education of secondary school and college/above had a significant association. Reduce or increase the odds of NNM ?

- Line 303 – references ?

- Line 309- Male neonates had a significant…… Please start in new paragraph

Lien 310 -311- This might be due to the increased

- 311 birth weight, cesarean sections, and operative deliveries were significantly higher for males. Is this from your findings or other ? If other, you need to cite reference .

- Line 315 - show the proximate factors….. What are the proximate factors? I didn’t see it in the background or methods. You need to state what are the proximate factors

- Line 316 -318 - Therefore, this study identified some new variables were not identified by others study, such as paternal educational status, Family monthly income and being male neonate to fill the previously mentioned gaps. This is not true . other studies have also reported the association between NNM and education and family income. For instance, in the article ‘Incidence and determinants of neonatal near miss in south Ethiopia: a prospective cohort study’ the authors stated income was associated with NNM.

- The authors need to explain the limitation of the study

-

Reviewer #3: The article is important because the neonatal near miss concept must to be present as a tool for evaluating neonatal care. It is the first step in building management strategies to reduce mortality and long-term sequelae. In the conclusions, my suggestion for the authors, would work more in local y practical suggestions for improving and avoid some causes.

Unfortunatlly many and the main causes are more structural, as poverty or education , but some evidencies show the necesity for improve the quality of care as a early screening and manage multiple pregnancy and pregnancy-induced hypertension,

Reviewer #4: It is a very interesting study concerning an important issue such as neonatal near miss (NNM). I have some concerns which could improve the paper and become it more suitable por publication.

INTRODUCTION is too long, should be shortened to 4 or 5 paragraphs focusing in the main point, NNM. Even though the authors are using correct references for definition, some misunderstanding occurs when they cite totally different concepts such as SIDS (sudden infant death syndrome), that has to be erased. Some epidemiological numbers are repetitive.

METHODS are very correct, defining very well area of the study, study design and selection of cases and controls.

Statistical approach is adequate, including a sample size calculation that was respected to obtain an adequate power for the study.

It is interesting the concern with the transcultural adaptation of the questionaire, very appropriate.

The concept of NNM are absolutely correct, well utilized, data analysis adequate and ethical aspects were well considered and approached.

RESULTS and DISCUSSION are interesting, not difficult to understand, because the adjusted analysis identified factors that really could interfere in the occurence of NNM or neonatal death. Even though male sex is classicaly reconized as an significant risk factor, the explanations are speculations. A new literature search should better address this point.

In my opinion, the main point in which I disagree are the explanations for assisted vaginal deliveries (AVD) as a risk factors. WHO is working with a Task Force to increase and improve AVB use worldwide, and seems to be that the main difficulty, to indicate AVD in a more appropriate time, during second period of the labour. I would study better this point and rewrite explanations. May be the professionals are not skilled enough, may be the mothers are frightened by the indications that happen lately in worse conditions for the fetuses and so on.

All at all is a good study, that deserves to be published correcting the main issues.

I suggest a professional English edition.

6. PLOS authors have the option to publish the peer review history of their article (what does this mean?). If published, this will include your full peer review and any attached files.

Reviewer #1: **Yes: **Wubet Alebachew Bayih

Reviewer #2: No

Reviewer #3: **Yes: **Suzanne J Serruya

Reviewer #4: No

---

## [Author Response · Author response to Decision Letter 0]

17 Jun 2022

Response to Reviewers’ 

Title: “FACTORS ASSOCIATED WITH NEONATAL NEAR MISS AMONG NEONATES ADMITTED TO PUBLIC HOSPITALS IN DIRE DAWA ADMINISTRATION, EASTERN ETHIOPIA: A CASE-CONTROL STUDY." [Manuscript ID PONE-D-21-33319]

To: The editor-in-chief, PLOS ONE

From: Authors 

Subject: Revision of the manuscript 

Dear Sir/ Madam,

We hope everything is fine. We appreciate and thank to the academic editor and reviewers for investing their time and energy to review and make comments on our manuscript. It is with great pleasure to receive the invaluable and constructive comments for our manuscript. 

As per your request for a separate cover letter, the comments with their point-by-point responses are put below here. In addition, the detailed changes made are highlighted in the “revised manuscript with track changes” by activating the “track changes” feature to easily identify the changes/ improvements. Moreover, the manuscript without track change is prepared. We accepted and tried to incorporate all of the comments provided. Therefore, we are kindly requesting you to review our revised manuscript; especially the “manuscript without track change”. If there is/are any unaddressed issue you welcome, we are ready to accept your comment.

No Comments Responses

Editor 

1. Please ensure that your manuscript meets PLOS ONE's style requirements, including those for file naming. Thank you for your concern. It meets PLOS ONE’s style requirements.

2. You indicated that you had ethical approval for your study. In your Methods section, please ensure you have also stated whether you obtained consent from parents or guardians of the minors included in the study or whether the research ethics committee or IRB specifically waived the need for their consent. Thank you for your concern. We have received ethical approval. For minors we have taken consent from their guardians after taking assent from the minors. This is also approved by the ethical review committee. 

 Review Comments to the Author 

 Reviewer 1 

- Some of the language use errors which the authors made consistently include.

Eg. 

a. Card review…… instead of chart review

b. Bivariate and multivariate ….. instead of bivariable and multivariable Thank you for the comment. We addressed it and incorporated in the manuscript.

1. Background of the abstract doesn’t clearly show the existing numerical burden of the problem in Dire Dawa town or other regional states of Ethiopia even. Generally, burden of neonatal near miss should be stated numerically followed by the objectives showing the research gap the authors would like to address. Thank you for the comment. Dear, as you know there is restrictions of word in abstract, we are unable to include as your comment. However, it is possible to find main body of the document.

2. The fourth sentence of the background that reads as ‘However, little studies were done in other areas, but they failed in identifying the proximate factors and not done in Dire Dawa Administration’ should be rewritten as: Though there were few prior studies in other regions, they failed in identifying the proximate factors. Besides, there has been no prior study in the study area. Thank you for the comment. We did per the comment and incorporated in the main document.

3. Methods of abstract should include sampling technique, measurement of neonatal nearmiss and type of data collection tool (adapted or adopted) Thank you for the comment. We did all comments per the comment and incorporated in the main document.

4. Result of abstract: please include response rate in the first sentence of the result section Thank you for the comment. We did per the comment and incorporated in the main document.

5. Data collection tools, procedures, and quality control: It would be more self explanatory and easily understandable if the authors showed pictorial presentation (flow chart) of the sampling procedure to reach a response rate of 100%. Please upload your data collection tools as additional file than annexing it in the manuscript. Kindly state the exact number of cases and controls rather than mentioning the routine statistics i.e. 5% of the sample size. Thank you for the comment. We did per the comment and incorporated in the main document. Also, the too is uploaded.

6. Ethical consideration: kindly have a separate upload of the ethical letter as an additional file. Moreover, what beneficent actions did the authors provide the mothers and the facilities in return for the interviews and chart reviews? Thank you for the concern. We did per the comment and incorporated in the main document. Also, it is uploaded.

There is no direct benefit given to mother or the facility except the consent receiving from both the facility and mothers. The benefit of both of them might be future interventions of the finding of the study.

7. How were you able to measure monthly income because you haven’t planned principal component analysis? Thank you for the concern. We know it should be assessed with principal components, but we simply asked them their estimated monthly incomes. This might be our limitation.

We wonders your comment and have a great respect. For any unaddressed comments, always we are ready/voluntary to correct again. So, please common in any inquiry. Thank.

 Reviewer 2 

 - Line 21- The near-miss cases are subject to factors…… can you say it neonatal near miss instead of near-miss since there are other miss e.g maternal near miss. Thank you for the concern. We did per the comment and incorporated in the main document.

 - I do not think by chance is appropriate word to use. Please get rid of the word ‘chance’ Thank you for the concern. We did per the comment and incorporated in the main document.

 - Line 29 – say medical record instead of ‘card’ Thank you for the concern. We did per the comment.

 - Line 30 – Multivariate is not the correct word , replace it by ‘multivariable analysis’ Thank you for the concern. We did per the comment.

 - Line 31-32 -the authors need to re-write the sentence Thank you for the concern. We did per the comment.

 - Line 33 – no need of mentioning in the abstract about p-values. Please take it out from the abstract. Thank you for the concern. We did per the comment.

 - Line 35 -39 – can you make the C.Interval to two digits rather than three. I don’t recommend stating adjusted ratios and CI in the abstract. It would be better if you could mention the factors associated with NNM without AOR. Thank you for the concern. We did for the 2 digit decimals. However, in most literatures it is advisable to indicate the measure of association (AOR and CI) in abstract since it is the short cut of the whole results. Unless it will overlaps with concussion.

 - Line 41 to 43 -seems repetition of results. I would recommend to state in other ways. Thank you for the comment and we do have respect for your comment. However, the detailed presented in the result section, the conclusion need to be stated by putting all factors as it indicated.

 - Better to create job opportunities, improving education, income generation. How? In any particular way? Thank you for the comment. For the sec of word number, we added the details in the main document conclusion.

 - …… minimizing instrumental delivery. How? Thank you for the comment. For the sec of word number, we added the details in the main document conclusion.

 - I would start the background with paragraph 4 ‘Worldwide, about 3.6 million neonates are estimated to die in the first 4 weeks of life every year and the majority continue to die at home, uncounted.’ The authors need to change the order of the paragraphs in order to make it easy for readers Thank you for the concerns and we do have a respect to your comment. But, background need to be start with outcome variable definition (introducing what mean your outcome variable), then showing the magnitude of the problem. So, as you seen on the first paragraph it is the definitions of the outcome variable; second paragraph, the criteria to diagnose (identify) the cases and the third paragraph shows the venerability period where the risk of neonatal death will occur. Then the rest paragraphs are showing the magnitude of the problem from world to local area. Therefore, we hope that the flow of the paragraphs are in correct order. Any way, if still is not convincing we will try to rearrange it. Thank you again.

 - It needs language revision Thank you for the concern. We tried to revise it.

 - Can you add more information about study facility such as availability of ICU units, paediatrics admission and number of deliveries per year ? Thank you for the concern. We have added the information in the main document.

 - Line 111 -12 – not clear Thank you for your detailed revision. We were missed some phrases and now it is complete and clear.

 - Line 116 – 120 – It is long sentence. Can you make it two sentences? Thank you for the comment. We have made accordingly in the main document.

 - Line 121 – 127 – Repetition of background information. It would be good if the authors mention the criteria of NNM only in methods part. Please take it out from the background. Thank you for the comment. We accepted the comment and we removed from the background.

 - Line 137- please include reference. Thank you for the comment. We added the reference.

 - Line 148- ‘Data were collected by 6 midwives who have experience in maternal and neonatal care’. Are the midwives hospital staff? Thank you for the comment. The data collectors are not the hospital staffs. They were recruited from other area. This also know indicated in main document.

 - Line 150 -153- Please make it two sentence. Thank you for the comment. We accepted the comment and we made accordingly.

 - Can you include reference number for ethics approval? Thank you for the comment. We have include reference number for ethics approval (ም/ማ/አ/ም/ፕ300/879/2013).

 - ‘This might be due to that the adverse effects of occupational stress on fetal

growth and development. Do you have reference for this? Can we say private /non-governmental employment increase the risk of adverse outcomes? Thank you for the comment. We linked this sentence with the second sentence of the same paragraph that have reference number 15. As we know Private/non-governmental organizations are business-oriented organizations most of the time workers are more loaded than governmental works (governmental organization control is weaker than the private). 

 - Line 302 _ having paternal education of secondary school and college/above had a significant association. Reduce or increase the odds of NNM ? Thank you for the concern. Under the subtitle of factors associated with the neonatal near miss, we clearly indicated that as ‘Having paternal education of secondary school (9-12) and college/above were less likely to develop NNM cases compared to those with no formal education’.

 - Line 303 – references ? Thank you for the comment. We added the refrence.

 - Line 309- Male neonates had a significant…… Please start in new paragraph Thank you for the comment. We did it.

 -Lien 310 -311- This might be due to the increased Thank you for the comment. Since it was our personal suggestions, we removed from the document.

 - 311 birth weight, cesarean sections, and operative deliveries were significantly higher for males. Is this from your findings or other ? If other, you need to cite reference . 

 - Line 315 - show the proximate factors….. What are the proximate factors? I didn’t see it in the background or methods. You need to state what are the proximate factors Thank you for the comment. Since we were inappropriately used. We removed the word proximate from the document.

 - Line 316 -318 - Therefore, this study identified some new variables were not identified by others study, such as paternal educational status, Family monthly income and being male neonate to fill the previously mentioned gaps. This is not true . other studies have also reported the association between NNM and education and family income. For instance, in the article ‘Incidence and determinants of neonatal near miss in south Ethiopia: a prospective cohort study’ the authors stated income was associated with NNM. Greatly we want to thanks you for your detailed review. We were missed this reference. Know we added this reference and corrected per the comment in the document.

 - The authors need to explain the limitation of the study Thank you again for your comment. The last sentence of the discussion section is the limitation of the study.

We wonders your comment and have a great respect. For any unaddressed comments, always we are ready/voluntary to correct again. So, please common in any inquiry. Thank.

 Reviewer 3: 

 The article is important because the neonatal near miss concept must to be present as a tool for evaluating neonatal care. It is the first step in building management strategies to reduce mortality and long-term sequelae. In the conclusions, my suggestion for the authors, would work more in local y practical suggestions for improving and avoid some causes.

Unfortunatlly many and the main causes are more structural, as poverty or education , but some evidencies show the necesity for improve the quality of care as a early screening and manage multiple pregnancy and pregnancy-induced hypertension, Thank you for your all the insight you shown us on the our work. Directly or indirectly we addressed the improvement of the service quality through a early screening and manage multiple pregnancy and pregnancy-induced hypertension, and minimizing instrumental delivery through careful (expertise) assessment of the indications our finding related recommendations in locally practical recommendations. This to maintain our result and recommendation agreement.

Great thanks again. If not addressed you well come again, we are ready to accept your comment.

 Reviewer 4: 

 INTRODUCTION is too long, should be shortened to 4 or 5 paragraphs focusing in the main point, NNM. Even though the authors are using correct references for definition, some misunderstanding occurs when they cite totally different concepts such as SIDS (sudden infant death syndrome), that has to be erased. Some epidemiological numbers are repetitive. Thank you for your constructive comment. We have tried to minimize the paragraph by removing some overlapping concepts in the document.

 METHODS are very correct, defining very well area of the study, study design and selection of cases and controls. Thank you for your constructive support.

 Statistical approach is adequate, including a sample size calculation that was respected to obtain an adequate power for the study.

It is interesting the concern with the transcultural adaptation of the questionaire, very appropriate.

The concept of NNM are absolutely correct, well utilized, data analysis adequate and ethical aspects were well considered and approached. Thank you for your constructive support.

 RESULTS and DISCUSSION are interesting, not difficult to understand, because the adjusted analysis identified factors that really could interfere in the occurence of NNM or neonatal death. Even though male sex is classicaly reconized as an significant risk factor, the explanations are speculations. A new literature search should better address this point. Thank you again for the constructive comment. We only one reference to support for male neonate findings. However, we removed one sentence that might speculate.

 In my opinion, the main point in which I disagree are the explanations for assisted vaginal deliveries (AVD) as a risk factors. WHO is working with a Task Force to increase and improve AVB use worldwide, and seems to be that the main difficulty, to indicate AVD in a more appropriate time, during second period of the labour. I would study better this point and rewrite explanations. May be the professionals are not skilled enough, may be the mothers are frightened by the indications that happen lately in worse conditions for the fetuses and so on. Thank you again for the constructive comment. This is might be because of we compared AVD with SVD. Any way we have taken directly your recommendation by removing our second sentence that describes the instrumental risk. 

 All at all is a good study, that deserves to be published correcting the main issues. Thank you again for the constructive comment.

 I suggest a professional English edition. Thank you again for the constructive comment. We have tried to rephrase it.

We wonders your comment and have a great respect. For any unaddressed comments, always we are ready/voluntary to correct again. So, please common in any inquiry. Thank.

Thank you,

Authors

---

## [Decision Letter · Decision Letter 1]

15 Aug 2022

FACTORS ASSOCIATED WITH NEONATAL NEAR MISS AMONG NEONATES ADMITTED TO PUBLIC HOSPITALS IN DIRE DAWA ADMINISTRATION, EASTERN ETHIOPIA: A CASE-CONTROL STUDY

PONE-D-21-33319R1

Dear Dr. Sintayehu,

We’re pleased to inform you that your manuscript has been judged scientifically suitable for publication and will be formally accepted for publication once it meets all outstanding technical requirements.

Kind regards,

Simone Garzon

Academic Editor

PLOS ONE

Additional Editor Comments (optional):

Reviewers' comments:

Reviewer's Responses to Questions

**Comments to the Author**

1. If the authors have adequately addressed your comments raised in a previous round of review and you feel that this manuscript is now acceptable for publication, you may indicate that here to bypass the “Comments to the Author” section, enter your conflict of interest statement in the “Confidential to Editor” section, and submit your "Accept" recommendation.

Reviewer #3: All comments have been addressed

Reviewer #4: All comments have been addressed

2. Is the manuscript technically sound, and do the data support the conclusions?

Reviewer #3: Yes

Reviewer #4: Yes

3. Has the statistical analysis been performed appropriately and rigorously? 

Reviewer #3: N/A

Reviewer #4: Yes

4. Have the authors made all data underlying the findings in their manuscript fully available?

Reviewer #3: Yes

Reviewer #4: Yes

5. Is the manuscript presented in an intelligible fashion and written in standard English?

Reviewer #3: Yes

Reviewer #4: Yes

6. Review Comments to the Author

Reviewer #3: The issue of neonatal near miss needs to gain visibility. If the health services are able to establish neonatal surveillance and train care teams, we would not only improve mortality indicators as well as the subsequent development of newborns. The results shown that the factors that affects

neonatal near miss are strongly related to social determinants such as maternal occupation or paternal education shows the importance of having sectoral policies and social programs to protect pregnant women.

Reviewer #4: I consider that the main issues have been addressed and the manuscript is suitable for publication. Neonatal near miss is an important concept developed in order to assess quality of infantile-maternal services/maternities, but also permits investigators to compare units worldwide, i.e., has a valuable academic importance.

7. PLOS authors have the option to publish the peer review history of their article (what does this mean?). If published, this will include your full peer review and any attached files.

Reviewer #3: No

Reviewer #4: No

---

## [Editor Report · Acceptance letter]

19 Aug 2022

PONE-D-21-33319R1 

FACTORS ASSOCIATED WITH NEONATAL NEAR MISS AMONG NEONATES ADMITTED TO PUBLIC HOSPITALS IN DIRE DAWA ADMINISTRATION, EASTERN ETHIOPIA: A CASE-CONTROL STUDY 

Dear Dr. Sintayehu:

I'm pleased to inform you that your manuscript has been deemed suitable for publication in PLOS ONE. Congratulations! Your manuscript is now with our production department. 

Kind regards, 

on behalf of

Dr. Simone Garzon 

Academic Editor

PLOS ONE